# PI3Kγ Mediates Microglial Proliferation and Cell Viability via ROS

**DOI:** 10.3390/cells10102534

**Published:** 2021-09-24

**Authors:** Caroline Schmidt, Nadine Schneble-Löhnert, Trim Lajqi, Reinhard Wetzker, Jörg P. Müller, Reinhard Bauer

**Affiliations:** 1Center for Molecular Biomedicine, Institute of Molecular Cell Biology, Jena University Hospital, 07745 Jena, Germany; caroline.schmidt1983@gmail.com (C.S.); nadschneble@freenet.de (N.S.-L.); joerg.mueller@med.uni-jena.de (J.P.M.); 2Department of Neonatology, Heidelberg University Children’s Hospital, 69120 Heidelberg, Germany; Trim.Lajqi@med.uni-heidelberg.de; 3Department of Anesthesiology and Intensive Care Medicine, Jena University Hospital, 07747 Jena, Germany; REINHARD.WETZKER@med.uni-jena.de

**Keywords:** phosphoinositide 3-kinase γ, proliferation, cell viability, microglia, ROS, LPS, ATP

## Abstract

(1) Background: Rapid microglial proliferation contributes to the complex responses of the innate immune system in the brain to various neuroinflammatory stimuli. Here, we investigated the regulatory function of phosphoinositide 3-kinase γ (PI3Kγ) and reactive oxygen species (ROS) for rapid proliferation of murine microglia induced by LPS and ATP. (2) Methods: PI3Kγ knockout mice (PI3Kγ KO), mice expressing catalytically inactive PI3Kγ (PI3Kγ KD) and wild-type mice were assessed for microglial proliferation using an in vivo wound healing assay. Additionally, primary microglia derived from newborn wild-type, PI3Kγ KO and PI3Kγ KD mice were used to analyze PI3Kγ effects on proliferation and cell viability, senescence and cellular and mitochondrial ROS production; the consequences of ROS production for proliferation and cell viability after LPS or ATP stimulation were studied using genetic and pharmacologic approaches. (3) Results: Mice with a loss of lipid kinase activity showed impaired proliferation of microglia. The prerequisite of induced microglial proliferation and cell viability appeared to be PI3Kγ-mediated induction of ROS production. (4) Conclusions: The lipid kinase activity of PI3Kγ plays a crucial role for microglial proliferation and cell viability after acute inflammatory activation.

## 1. Introduction

Microglia, the self-renewing innate immune cells of the central nervous system (CNS), are the resident macrophages of the CNS. They are the key mediators of the innate immune response [1,2]. Furthermore, they control the neuronal patterning and wiring of the brain in early development and fulfill supportive functions for maintenance of tissue homeostasis, tissue support, neuroplasticity and neuroprotection due to their ability to surveil the microenvironment for alterations [3,4,5,6]. Therefore, surveillant microglia continuously scan their environment for chemoattractants [7,8]. Insults or infections affecting the CNS are accompanied by a series of parenchymal events including release of damage-associated molecular pattern molecules (DAMPs) from impaired tissue or invasion of pathogen molecular pattern molecules (PAMPs) such as lipopolysaccharide (LPS), which results in activation of microglia and their increased motility and phagocytic activities [1,9,10]. Activated microglia are a main player in clearing dead neural cells, removing infections and restoring neuronal functions but may also accelerate proinflammatory responses with potential exacerbation of brain injury [6,11].

Microglia represent tissue-specific macrophages with a lifelong renewal characterized by a slow turnover and a stable cell density throughout the lifetime [5]. However, microglia can register disturbances of their environment and change their behavior. The first response includes convergence of the microglial processes on the site of injury without cell body movement, establishing a potential barrier between the healthy and injured tissue [8]. Intensified environmental disturbance resulting in injury or pathology leads to a quick microglial immune response, often termed as “activation”, defined as any physical or biochemical changes occurring outside of the microglial homeostatic state. Microglial “activation” includes rapid proliferation, migration to the site of pathology, phagocytosis of cells and debris and production of cytokines and chemokines, necessary to stimulate microglia and other brain and immune cells [12,13].

Remarkably, among other features, the rapid switch from a very slow propagating self-renewing phenotype to a rapidly proliferating state is responsible for promoting microglia as the prominent immune-competent cells in the early hours after the onset of acute brain injury such as focal brain ischemia [14,15,16]. Although a plethora of pathological mechanisms in response to acute brain injury have been partly elucidated including the temporal profile of the occurrence and propagation of immune-competent cells, the understanding of the detailed molecular events lacks precision [17,18].

PI3Kγ is the sole subclass IB member of class I phosphoinositide 3-kinases (PI3Ks), characterized by activation via G protein β-γ subunits, which are usually derived from Gi-coupled receptors [19,20]. PI3Ks were originally characterized as signaling proteins which generate phosphatidylinositol 3,4,5-trisphosphate for subsequent protein kinase B/Akt activation via their lipid kinase activity [21,22,23]. Furthermore, our studies demonstrated an intimate interplay of PI3Kγ with cAMP signaling pathways via PI3Kγ-mediated activation of phosphodiesterases (PDE), leading to inhibition of inflammatory responses [9,24,25,26].

PI3Kγ was identified as the major PI3K catalytic isoform in primary myeloid cells, as these cells express at least 25-fold more PI3K than other isoforms [27]. We and others showed solid expression of PI3Kγ in primary microglial cells [9,28] and documented their crucial impact on the microglial immune response: directed migration, MMP-9 expression and phagocytic activity following systemic inflammation and focal brain ischemia/recirculation as well as modulation of innate immune memory [10,24,26,29,30,31].

What are the key signaling events controlling the molecular patterns for proliferation and cell viability in activated microglial cells? Here, we addressed this question.

We discovered that mice with a loss of lipid kinase activity show an impaired proliferation and cell viability of microglia. Specifically, by using genetic and pharmacologic approaches, we demonstrate that both DAMP ATP and the prototypical PAMP LPS activate microglial proliferation and cell viability in a PI3Kγ-dependent manner.

## 2. Materials and Methods

### 2.1. Antibodies

Monoclonal anti-mouse PI3Kγ antibody was produced in our facility [23]. Other antibodies were obtained from Cell Signaling Technologies (Danvers, MA, USA) (PCNA, #2586); Sigma Aldrich (Steinheim, Germany) (β actin, #A5441); Cell Signaling (Danvers, MA, USA) (phospho-Akt (Ser473), #9271; Akt, #9106; phospho-p44/42 MAPK (ERK1/2) (Thr202/Tyr204), #9107; p44/42 MAPK (ERK1/2), #9107); and Abcam (Berlin, Germany) (Iba1 (#ab5076) and Ki67 (ab16667)). Secondary antibodies Alexa 488 donkey anti-rabbit (#A21206) and Alexa 568 donkey anti-goat (#A11057) were purchased from Invitrogen (Carlsbad, CA, USA), whereas HRP-coupled anti-rabbit and anti-mouse antibodies were purchased from KPL (Weden, Germany).

### 2.2. Inhibitors

The PI3Kγ inhibitor AS605240 (1 µM) was obtained from Alexis (Lausen, Switzerland). Inhibitors for PI3Kα (A66; 300 nM), for PI3Kβ (TGX221; 200 nM) and for PI3Kδ (IC87114; 2 µM) were purchased from the Baker Heart Research Institute (Melbourne, Australia). Inhibitor A66 (PI3Kδ) was obtained from Symansis (Auckland, New Zealand). Cellular and mitochondrial ROS inhibitors N-acetylcysteine (NAC) and 4,4′-diisothiocyanatostilbene-2,2′-disulfonic acid disodium salt (DIDS) were obtained from Sigma (Saint Louis, MO, USA).

### 2.3. Other Reagents

Lipopolysaccharide (LPS), adenosine triphosphate (ATP), 1-(4,5-Dimethylthiazol-2-yl)-3,5-diphenylformazan (MTT), propidium iodide (PI) and X-Gal were purchased from Sigma Aldrich (Steinheim, Germany). NGF, TNF-α, M-CSF and amyloid beta were obtained from ImmunoTools (Friesoythe; Germany) and Abcam (Cambridge, UK). H2DCFDA (DCF) for cellular ROS measurement and MitoSOX™ Red for mitochondrial superoxide measurements were obtained from Invitrogen (Carlsbad, CA, USA).

### 2.4. Animals

PI3Kγ knockout mice (PI3Kγ KO) [32] and PI3Kγ kinase-dead mice (PI3Kγ KD, mice carrying a targeted point mutation in the PI3Kγ gene causing loss of lipid kinase activity) [33] were on the C57BL/6J background for more than ten generations. Consequently, age-matched C57BL/6 mice were used as controls. The animals were maintained with 12 h light/dark cycles with free access to food and water. Ambient temperature was 29 ± 1 °C during the entire experimental period. The animal procedures were performed according to the guidelines from Directive 2010/63/EU of the European Parliament on the protection of animals used for scientific purposes. Experiments were approved by the Thuringian State Office for Food Safety and Consumer Protection. Efforts were made to reduce the number of animals used and their suffering. All surgeries were performed under appropriate anesthesia (see below).

### 2.5. Primary Microglia Cell Isolation and Cultivation

Primary microglia cells were isolated from C57BL/6J wild-type, PI3Kγ KO and PI3Kγ KD neonatal mice. The isolation was performed on the cerebral cortex (P1-3), as described previously [9,29,34]. Cells were cultivated in co-culture with astrocytes for 14 days at 37 °C and 5% CO_2_ in DMEM High Glucose containing 10% FCS, 1% penicillin/streptomycin and 1% amphotericin B in T75 cell culture flasks. After 7 days, 50% of the medium was replaced by fresh medium. Fourteen days after cultivation, microglial cells were separated from astrocytes by adding 1 mL PBS (containing 1 mM EDTA) by careful shaking. After harvesting, microglial cells were seeded into appropriate well plates.

### 2.6. SDS-PAGE and Immunoblotting

For quantification of protein expression and phosphorylation, cells were seeded into 6-well plates and incubated at 37 °C (5% CO_2_). After becoming adherent, cells were incubated overnight in DMEM medium without FCS and treated for 24 h with LPS/ATP and/or pretreatment for 60 min with inhibitors as indicated. Thereafter, cells were suspended in RIPA lysis buffer composed of 50 mM Tris/HCl pH 8.0, 150 mM NaCl, 1.0% (*v*/*v*) NP-40, 0.5% (*v*/*v*) deoxycholate, 0.1% (*w*/*v*) SDS, 100 µg/mL Pefa-Block, 1 µg/mL pepstatin, 10 µM sodium vanadate and 1 µg/mL leupeptin. Supernatants were mixed with 5× protein sample buffer (5% SDS, 33% glycerol, 5% β-mercaptoethanol) and heated to 95 °C for 5 min. Protein samples were separated on 10% polyacrylamide gels, transferred to a PVDF membrane and subsequently treated with indicated antibodies. Protein bands were detected by enhanced chemiluminescence reaction using an LAS4000 camera (Fuji Photo Film Co., Tokyo, Japan). Quantification of the protein bands on the membrane was conducted using the FujiFilm Multi Gauge Ver. 3.0 software (Fuji Photo Film Co., Tokyo, Japan).

### 2.7. In Vitro Cell Proliferation and Cell Viability Assessment

An MTT assay was performed as described [35]. Microglial cells were seeded in 96-well plates and stimulated with different concentrations of LPS (0, 10, 100 or 1000 ng/mL) or ATP (0, 10, 100 or 1000 µM) over 6 days as indicated. Fresh medium with stimulants was added every day. After indicated time points, 10 µL MTT (3-(4, 5-dimethylthiazolyl-2)-2, 5-diphenyltetrazolium bromide) solution (0.5 mg/mL in PBS) was added into each well, mixed by careful shaking and incubated for 4 h at 37 °C, 5% CO_2_. Subsequently, the converted MTT dye was solubilized by adding ethanol. Absorbance (λ = 570 nm) was measured after 30 min.

For propidium iodide (PI) staining, microglia were seeded into 6-well plates and stimulated with different doses of LPS over 6 days. After stimulation, cells were harvested by adding PBS (containing 2 mM EDTA), resuspended in PBS and subsequently fixed with 70% ethanol, 60 min at 4 °C. Thereafter, cells were centrifuged for 5 min at 2000 rpm, 4 °C. For permeabilization, the pellet was washed two times with PBS (containing 0.5% Triton X-100) and again centrifuged at 1600 rpm, 5 min, 4 °C. For PI staining, the cells were resuspended in PBS buffer containing 100 µL RNase (200 µg/mL) and incubated for 20 min at 37 °C. Afterwards, 100 µL of PI staining solution (150 µg/mL in PBS) was added and incubated for 15 min at 37 °C in the dark. Cell cycle analysis was conducted by flow cytometry measurement using FACS Canto (BD, Heidelberg, Germany).

### 2.8. Senescence Assay

For identifying senescent cells, senescence-associated β-galactosidase activity (SA-β-gal) staining was performed. Primary microglia were seeded into 12-well plates and stimulated with LPS for 6 days. After stimulation medium was removed, cells were washed once with PBS. Immediately thereafter, cells were fixed for 5 min at room temperature using 0.2% formaldehyde/0.02% glutaraldehyde. Fixed cells were washed again two times with PBS, and finally, 1.5 mL staining solution (X-Gal) was added to each dish. Cells were left in a dry incubator at 37 °C overnight. While the staining solution was still on the culture dish, cells were checked under a microscope (10× magnification) for the development of a blue color. Five pictures were taken in random fields, and senescent cells were counted. Senescent cells were calculated as a percentage of blue-positive SA-β-gal cells compared to the total number of cells in the respective fields.

### 2.9. Measurement of ROS

Measurement of reactive oxygen species in cells was performed using the H2DCFDA assay as described [29,31]. Microglial cells were seeded into white clear-bottom 96-well plates (30,000 cells/well). After becoming adherent, LPS, ATP and inhibitors were added at indicated time points. For measurement of intracellular ROS, cells were transferred to 200 µL H2DCFDA solution (50 mM, 1:1000 in 10 mM HEPES/CaCl_2_ solution) 24 h after stimulation and incubated for 20 min at 37 °C. Thereafter, each well was carefully washed two times with HEPES/CaCl_2_ solution. Converted fluorescence-active 2′,7′-Dichlorfluorescein (DCF) was quantified using a plate reader (Tecan infinite 2000, excitation at 485 nm, emission at 535 nm). Measurement of mitochondrial ROS was performed by following the manufacturer’s protocol (MitoSOX™ Red, Invitrogen, Carlsbad, CA, USA).

### 2.10. In Vivo Microglial Proliferation Assay

Experiments were performed on male 10- to 14-week-old wild-type, PI3Kγ KO and PI3Kγ KD mice (4 to 5 mice per group). To investigate the effect of PI3Kγ KO as well as PI3Kγ KD on microglial proliferation, an in vivo wound healing experiment was performed. Mice were anesthetized by intraperitoneal injection of a cocktail comprising midazolam (5 mg/kg), fentanyl (0.05 mg/kg) and medetomidine (0.5 mg/kg) and positioned in a stereotaxic apparatus (Stoelting, Wood Dale, IL, USA). Mice were then placed on a homeothermic heat blanket to maintain normal body temperature during surgery. The skull was exposed by a skin incision, and small burr holes were drilled through the skull. Using a micromanipulator, a focal stab injury was performed by gentle insertion of a stainless-steel pin (diameter 0.25 mm) into the brain (coordinates: 1.5 mm rostral the bregma, 1.5 mm lateral from midline on each side, 2 mm in depth) [36,37]. The pin was kept in place for five minutes and then removed. The burr holes were covered with bone wax, and the animals were returned to their cages. Forty-eight hours later, mice were deeply anesthetized and perfused with 4% PFA in phosphate buffer by cardiac puncture via the left ventricle. Brains were removed immediately after fixation, postfixed for 1 day in 4% PFA at 4 °C and embedded in paraffin. For quantitative immunohistochemical analysis of in vivo proliferation, coronal sections (6 µm) of the brain region containing the focal stab injury were prepared and thoroughly washed with 0.3% Triton X-100 containing PBS (shaking for 10 min, 10 times). Sections were stained with anti-Iba1 antibody (1:250) overnight to visualize microglia cells and anti-Ki67 (1:200) for 2 h at room temperature to visualize proliferating microglial cells. Evaluation of proliferating microglial cells was performed under a fluorescence microscope (Olympus BX61). To determine the cell number of microglia in the peri-region of the injury site, the total number of Iba1-positive cells was counted in each image. To analyze proliferating microglia, Iba-1/Ki67-double positive cells were counted in every image. Indices of proliferated cells were determined by the ratio of Iba-1/Ki67-positive cells to the total number of Iba1-positive cells.

### 2.11. Data Analysis and Statistical Procedures

Statistical analysis was performed using SigmaPlot Software, version 14.5 (Inpixon GmbH, Düsseldorf, Germany). Data are reported as mean + standard error of the mean (SEM). Comparisons between groups were conducted with one-way, two-way or three-way analysis of variance, as appropriate. If the normality test failed, Kruskal–Wallis one-way analysis of variance on ranks was used. Post hoc comparisons were conducted with the Holm–Sidak test or Dunn’s method, as appropriate. A *p*-value of <0.05 was considered to be statistically significant.

## 3. Results

### 3.1. Dose-Dependent Induction of Microglial Proliferation

To study the activation-induced proliferation and cell viability of primary microglia, cells were purified from neonatal mouse cerebral cortices of newborn C57BL/6J wild-type mice. After co-cultivation with astrocytes, microglial cells were incubated with different doses of LPS as a PAMP or ATP as a DAMP for 6 days. Both activation inductors resulted in a dose-dependent alteration in cell viability. An amount of 100 ng/mL LPS or 100 µM ATP resulted in a significant elevation in cell viability. The higher concentration of both decoys impaired microglial viability (Figure 1A).

### 3.2. Microglial Proliferation Is Mediated by PI3Kγ

To investigate the role of particular phosphoinositide 3-kinases on proliferation, primary microglia were pretreated with specific inhibitors targeting the lipid kinase activities of PI3Kα (by kinase inhibitor A66), PI3Kβ (by kinase inhibitor TGX-221), PI3Kγ (by kinase inhibitor AS605240) or PI3Kδ (by kinase inhibitor IC87114) and subsequently incubated with LPS (100 ng/mL) or ATP (100 µM) for 6 days, each. As a fixed amount of LPS or ATP was used for microglia activation, differences delivered by the MTT assay mainly reflect differences in cell number after incubation, i.e., effects of proliferation. While incubation with kinase inhibitors targeting PI3K isoforms α, β and δ did not affect microglial proliferation compared to solely LPS- or ATP-treated cells, inhibition of PI3Kγ diminished microglial viability to a level similar to unstimulated cells (Figure 1B). Thus, these data identify PI3Kγ as an essential regulator of the LPS- as well as ATP-mediated viability of murine primary microglia.

To verify the effects of PI3Kγ signaling on microglial viability, we analyzed dose-dependent LPS and ATP administration on the microglial viability of cells derived from PI3Kγ-deficient mice (KO) and mice expressing a lipid kinase-inactive (kinase-dead, KD) PI3Kγ protein. The data reveal that PI3Kγ-mediated stimulation of microglial viability is lipid kinase-dependent because PI3Kγ KO microglia as well as PI3Kγ KD microglia showed no dose-dependent effects of cell proliferation after LPS and ATP administration (Figure 1C). Furthermore, LPS administration resulted in suppressed viability of PI3Kγ-modified microglial cells (Figure 1C).

To gain further insight into the activation profile of microglial viability, primary cells were additionally stimulated with the inflammatory cytokines nerve growth factor (NGF), macrophage colony stimulating factor (M-CSF) or tumor necrosis factor α (TNFα) as well as the DAMP β-amyloid known to modulate microglial viability. While the treatment of wild-type microglia with NGF or β-amyloid did not show any effect on their viability, M-CSF or TNFα had a slight but insignificant effect on the viability of wild-type microglia. In contrast, microglia derived from PI3Kγ KO and PI3Kγ KD mice did not respond to the PAMP, DAMP or inflammatory cytokines under consideration, indicating PI3Kγ-mediated proliferative activation by its lipid kinase activity (Figure 1D).

Since the proliferating cell nuclear antigen (PCNA) has been used to validate cell proliferation [38], we analyzed the cellular level of the protein in the three PI3Kγ genotypes of microglia. While PCNA increased strongly upon prolonged incubation of WT microglia with LPS, the protein expression of PCNA was reduced in microglia with gene-modified PI3Kγ variants (Figure 1E and Appendix A). Interestingly, intracellular signaling analysis revealed an upregulation of microglial PI3Kγ proliferation owing to LPS stimulation together with an activation of the Akt and MAPK pathways, which were blunted in PI3Kγ-deficient as well as PI3Kγ KD microglial cells (Appendix A). This observation indicates a positive feedback stimulation on PI3Kγ expression.

To elaborate on the impact of PI3Kγ on microglial proliferation within the intact brains, a focal wound injury was provoked by introducing a small needle in the cortical area of the adult mice genotypes under consideration. Proliferating microglia were quantified by counting Iba-1/Ki67-double positive cells. The data reveal a markedly higher proliferation rate in the peri-region of the injury site of brains derived from wild-type mice compared to the same brain regions derived from PI3Kγ KO and PI3Kγ KD mice (Figure 2).

### 3.3. PI3Kγ Improved Viability in LPS-Activated Microglia

To gain further insight into the physiological consequences of LPS-mediated activation of microglia, we characterized the induction of senescence and apoptosis during in vitro incubation of purified primary microglia derived from WT and PI3Kγ KO mice. At day 1 post-seeding of cells, the rate of senescence-associated β-gal-positive cells was about 20% irrespective of genotype and LPS stimulation (Figure 3A). After 6 days, almost all cells became SA-β-gal-positive in the absence of LPS stimulation. Treatment of WT microglia with LPS reduced the induction of β-galactosidase, indicating suppression of senescence. This effect could not be observed in microglia derived from PI3Kγ KO mice (Figure 3A).

To further judge the induction of apoptosis, the induction of cumulative apoptosis was analyzed by detection of DNA fragmentation (i.e., induction of Sub-G1 cells) using flow cytometry. After one day, a low rate of Sub-G1 cells could be detected. Prolonged incubation of microglia induced the rate of Sub-G1 cells containing fragmented DNA in a PI3Kγ-dependent manner. On day 6, half of the cell population contained fragmented DNA in unstimulated wild-type microglia as well as LPS-stimulated PI3Kγ KO microglia (Figure 3B). While LPS stimulation induced a marked reduction in apoptotic WT microglia, a similar rate of apoptosis was observed in KO microglia incubated with LPS. In addition, at day 6, about 2/3 of the LPS-treated WT microglia were in the S or G2 phase (data not shown). These data indicate that—aside from the induction of proliferation—PI3Kγ mediated anti-apoptotic activation in response to LPS treatment (Figure 3).

### 3.4. PI3Kγ Mediates Microglial Proliferation via ROS Production

There is sufficient evidence that microglia, particularly in aged mice, undergo LPS-mediated ROS production (recently reviewed in [39]). Furthermore, Mander et al. showed that ROS act as second messengers in microglial activation and proliferation [40]. Accordingly, we addressed the question concerning whether PI3Kγ controls microglial ROS generation and implements—via this signaling transduction path—its function for microglial viability and proliferation. Therefore, to examine the role of class I PI3K isoforms in the control of ROS generation, wild-type microglia were pretreated with inhibitors specifically inhibiting the different lipid kinase activities of the four PI3K isoforms. While inhibition of PI3Kα, PI3Kβ or PI3Kδ did not affect the induction of cellular ROS production compared to the mock control, abrogation of PI3Kγ kinase activity significantly diminished cellular ROS production down to the levels observed in the untreated cells (Figure 4A,B). Thus, it can be concluded that PI3Kγ specifically mediates LPS- as well as ATP-mediated NOX-dependent ROS production.

To investigate the molecular pattern of cellular ROS induction, proliferation agonists NGF, M-CSF, TNFα and β-amyloid were used besides LPS and ATP. While LPS resulted in significant induction of cellular ROS production in wild-type microglia, other stimuli did not (Appendix A). Furthermore, the stimulatory effects of the agonists for cellular ROS production were blunted in cells derived from PI3Kγ KO as well as PI3Kγ KD mice.

Next, we verified the dose dependency of cellular and mitochondrial ROS production in microglia stimulated by LPS and ATP (Figure 4C). Intriguingly, the same dosage was potent enough to induce the maximum cellular ROS production, as it was found for the maximum viability. As expected, LPS- as well as ATP-mediated induction of ROS formation remained unchanged in microglia derived from PI3Kγ KO or PI3Kγ KD, irrespective of the concentration used. The fluorogenic dye MitoSOX specifically targeted to mitochondria in live cells was used to measure the induction of mitochondrial ROS production in response to microglia stimulation. While LPS stimulated MitoSOX oxidation over a wider spectrum of concentrations, a significant elevation in ATP-mediated mitochondrial ROS was merely observed at 100 µM ATP (Figure 4C). Since PI3Kγ KO and PI3Kγ KD microglia did not alter mitochondrial ROS production in response to LPS or ATP stimulation, it can be concluded that functional PI3Kγ triggered the signal transduction of cellular and mitochondrial ROS production.

### 3.5. PI3Kγ Controls Microglial Proliferation via ROS

To further characterize the PI3Kγ-mediated signal transduction pathways controlling the LPS- and ATP-mediated induction of microglial proliferation, we selectively abrogated NOX-dependent ROS by NAC and mitochondrial ROS by DIDS. The specificity of NAC and DIDS on NOX and mitochondrial ROS inhibition was shown by selective treatment.

Intriguingly, in wild-type microglia, the LPS- and ATP-induced increase in microglial proliferation was abrogated by blockade of cellular and mitochondrial ROS production. In contrast, PI3Kγ KO and PI3Kγ KD microglia did not alter their proliferative activity under the same condition under consideration (Figure 4D).

To verify the efficacy of the used inhibitor dosage to abrogate the LPS- or ATP-induced stimulation of cellular or mitochondrial ROS production, a selective treatment was performed on microglia derived from all three mouse genotypes under consideration, and the amount of ROS production was assessed. While LPS-induced cellular and mitochondrial ROS production in wild-type microglia was completely abrogated by the respective inhibitors, ATP-induced mitochondrial ROS production was maintained despite the NAC treatment. Furthermore, PI3Kγ KD microglia exhibited increased mitochondrial ROS production in response to ATP stimulation (Appendix A). In addition, microglia derived from PI3Kγ mutant mice did not show any relevant alterations in LPS- or ATP-stimulated ROS production as well as in the related inhibition of cellular or mitochondrial ROS production (Appendix A).

Taken together, our data reveal that PI3Kγ plays a central role in DAMP- or PAMP-mediated activation of microglial viability and proliferation.

## 4. Discussion

Microglia are resident macrophages of the brain contributing to the innate immune response of the CNS. They become activated by infection agents, especially their conserved molecules known as PAMPs, or by danger signals and cellular debris in response to injury. Selective receptors trigger the information to be communicated with the cell physiological functions such as directed migration to regions of injury, infection proliferation or induction of phagocytosis [26,36,41,42,43]. Here, we elucidated the role of PI3Kγ in PAMP- or DAMP-mediated microglial activation, viability and proliferation. By using genetic and pharmacological approaches, we demonstrate that PI3Kγ—and no other phosphoinositide 3-kinases—regulates microglial viability and proliferation induced by LPS or ATP. Molecular pattern-induced ROS—both in the cytosol and mitochondria—are the prerequisite for the suppression of apoptosis and induction of microglial proliferation. This fulfills the key function of microglia as the front line of the brain’s immune response to microbial pathogens or injury. Intriguingly, functional PI3Kγ kinase activity is an important contributor to microglial activation in response to brain injury. The comparative analysis of microglia derived from mice expressing wild-type PI3Kγ, PI3Kγ-deficient mice and mice with cells expressing the KD mutant of PI3Kγ clearly shows the eponymous enzymatic activity of PI3Kγ as an essential mediator of ROS production and cell proliferation [32,33].

Our study uncovers the inhibition of PI3Kγ lipid kinase activity as a mechanism causing suppression of the microglial proliferation of LPS or ATP. The dose levels of the prototypical bacterial PAMP as well as the DAMP triggering robust microglial activation used in our in vitro systems were based on our previous work [10,24,29,44]. Nevertheless, it should be recognized that relevant cellular immune responses can be induced—at least in naïve microglia—with much lower LPS concentrations. Indeed, LPS acts as an extraordinary PAMP able to stimulate host responses even in the femtomolar range [45,46,47]. Mechanistically, LPS-induced immune cell activation is mediated by TLR4 [48,49]. Receptor activation requires binding of an individual LPS molecule (LPS monomer) to the MD-2/TLR4 dimeric receptor [50] and dimerization of the LPS-MD-2-TLR4 ternary complex [51]. In turn, as few as 25 LPS MD-2-TLR4 complexes per cell can trigger measurable proinflammatory responses. Therefore, priming effects with ultra-low LPS doses are apparently potent enough to control intracellular training effects important for priming innate immune memory in neonatal microglia, as we have shown previously [31].

Based on the previous evidence of PI3Kγ-dependent control of different microglial cell functions including phagocytosis [9], microglial motility [10], memory-like inflammatory responses [29] and immunometabolic reprogramming for innate immune memory [30], a certain influence of PI3Kγ on microglial proliferation was anticipated. However, whereas PI3Kγ-dependent modulation of microglial phagocytic activity was clearly mediated by lipid kinase-independent stimulation of phosphodiesterase activity, leading to reduced intracellular cAMP with suppressed phagocytic activity [9], PI3Kγ lipid kinase activity was identified as an essential mediator of microglial migration and memory-like effects in microglia [10,29,30]. The present data verify that the lipid kinase activity of PI3Kγ is responsible for microglial proliferative activation and improved cell viability due to inflammatory stimulation. Whereas wild-type microglia containing functional PI3Kγ were able to respond to LPS and ATP in a dose-dependent manner, PI3Kγ-deficient microglia as well as microglia armed with KD PI3Kγ (but with preserved scaffold function) failed to do so [33]. We must consider that in the experimental series with dose-dependent stimulation of LPS-mediated cellular ROS production (Figure 4C), distinct quantitative differences occurred compared with the results obtained from comparable experimental conditions (presented in Figure 4A). Nevertheless, the main result—the dose-dependent cellular (and mitochondrial) ROS production in wild-type microglia induced by LPS—is clearly shown.

Taking into account that microglia are classified as the resident brain tissue macrophages and act as the first and main form of active immune defense in the CNS, microglia respond primarily to PAMPs and DAMPs—in addition to a great number of other mainly proinflammatory mediators—by the production of ROS [52]. There is evidence that ATP released from stimulated microglia induces ROS generation by NADPH oxidase activation via P2X7 receptor activation in an autocrine manner. This ROS generation was inhibited by a pan-PI3K inhibitor [53,54]. The presented data reveal that there is a dose dependency of LPS- and ATP-dependent ROS production in wild-type microglia, either of cellular or mitochondrial origin, which parallels comparable effects on microglial proliferation. The dose-dependent development of the microglial immune response is considered its critical feature. The differential reactions of microglia to increasing doses of PAMPs and DAMPs can be interpreted as “hormetic”, comprising sequential stimulation of pathogen-induced resistance and proliferation, followed by tolerance to pathogens and, finally, an irreversible state of cell damage [55,56,57]. Adaptive responses of immune cells to increasing doses of pathogens are typically accompanied by increasing metabolic exhaustion. Anabolic responses such as proliferation and pathogen resistance at higher doses turn to catabolic and repair responses during immune tolerance before cell death. Following this basic principle, we suggest that the decreased viability of microglia at high doses of LPS and ATP is embedded in the hormetic tolerance response scheme [58,59]. In contrast to resistance responses, tolerant immune cells do not attack pathogens and do not proliferate but perform repair reactions to limit damage induced by the microorganisms. Together, these considerations indicate at least a partial tolerance response pattern as a cause for the decreased proliferation at high LPS and ATP doses. In line with the findings reported herein, recently, we showed that naïve microglia display memory-like immune responses in a dose-dependent manner and revealed an essential role of PI3Kγ for signaling processes and cell functions in microglia [29].

Furthermore, our data reveal that prolonged LPS-induced microglial activation provoked a PI3Kγ-dependent partial suppression of replicative senescence (Figure 3A). This finding specifies, therefore, previous reports that cellular senescence, characterized by permanent cell cycle arrest and involving, among others, changes in the redox state and alterations in mitochondrial homeostasis [60,61], can be modified by different PI3K isotypes to regulate longevity as well as resistance to oxidative stress [62,63]. In addition, the anticipated anti-apoptotic effect of PI3Kγ could be verified as a result of prolonged microglial PAMP stimulation [64]. As we showed in Appendix A, LPS increased PI3Kγ expression at the protein level and led to enhanced Akt phosphorylation and hence activation. Consequently, different anti-apoptotic pathways are possibly initiated: (i) Inhibition of caspase 9, a protease crucial in the initiation of the apoptotic cascade, may be triggered [65]. (ii) Akt also phosphorylates the B cell lymphoma 2 (Bcl-2)-associated death promoter (BAD) and thus releases the anti-apoptotic proteins Bcl-2 and B cell lymphoma-extra large (Bcl-XL) [66,67]. (iii) Finally, inactivation and cytosolic retention of Forkhead-Box-Protein O (FOXO) leads to a block in the transcription of the Fas ligand (FasL) and thus intercepts ligand-induced apoptosis [68,69]. Thus, we could show that rapid microglial proliferation, which is provoked by appropriate LPS-induced microglial activation, is clearly accompanied by PI3Kγ-dependent enhanced cell cycle progression and suppression of apoptosis.

Intriguingly, an LPS- and ATP-dependent increase in cellular as well as mitochondrial ROS production in PI3Kγ KO and PI3Kγ KD microglia was missing. Therefore, the lipid kinase activity of PI3Kγ appears to be closely linked with inflammation-related microglial ROS production. Furthermore, PI3Kγ controls microglial proliferation via ROS, as evidenced by the pharmacological blockade of LPS-induced enhanced cellular as well as mitochondrial ROS production, which prevented a corresponding increase in the proliferation of wild-type microglia (Figure 4D). There is ample evidence that molecular pathways in microglia contributing to acute and chronic neurodegeneration undoubtedly involve inflammatory-mediated ROS production leading to oxidative stress and probably neurotoxicity by activation of NOX proteins. Notably, NOX2 is the most highly expressed NOX isotype transcript in human and mouse brain microglia [70,71]. However, ROS from both NOX and mitochondria contribute to inflammatory activation of microglia by acting as secondary messengers propagating inflammatory states [72]. Our data confirm and refine previous findings which claimed a PI3K-dependent control of the assembly of the oxidase complex in microglia [73]. Using a pan-PI3Ks blocker, Parvathenani et al. demonstrated an important role of class I PI3Ks in the regulation of various proteins of the NADPH oxidase complex in microglia [73]. With a genetic approach, we verified that the lipid kinase activity of PI3Kγ is crucial for cellular and mitochondrial ROS control. PI3Kγ occupies a prominent position within cell stress-related intracellular signaling networks of immune-competent cells. As such, PI3Kγ plays a remarkable role as the decisive isoform for immunological control of microglial proliferation, together with an intimate interplay with a range of other inflammation-related cell functions [74]. The increased expression of PI3Kγ induced by LPS and ATP in naïve microglia indicates its intimate involvement in intracellular inflammatory signaling. Indeed, PI3Kγ was identified as the major PI3K catalytic isoform in primary myeloid cells, as these cells express at least 25-fold more PI3Kγ than other isoforms [27]. Our data suggest PI3Kγ/AKT signaling as a mediator for the rapid switch from a very slow propagating self-renewing microglial phenotype to a rapidly proliferating phenotype.

Intriguingly, increased cellular and mitochondrial ROS production and increased microglial viability were likewise dependent on PI3Kγ enzymatic activity. A similar response pattern as for Akt phosphorylation was observed for phospho-ERK1/2. In wild-type microglial cells, LPS and ATP treatment provoked stimulation of Akt and ERK phosphorylation, whereas inhibition of these signals was observed after high-dose LPS priming. Both in response to LPS and ATP treatment, Akt and ERK phosphorylation was diminished in microglial cells deficient in PI3Kγ lipid kinase activity. These data confirm earlier work which showed an impact of PI3Kγ lipid kinase activity on microglial immune regulation via PI3Kγ/AKT signaling pathways [29].

To discuss methodological aspects, it was noted that the cell culture measurements of microglial viability and proliferation were performed using the MTT assay. The MTT assay has become one of the most popular techniques for the quantitative assessment of cell proliferation, viability and cytotoxicity, due to being a low-cost, fast and simple procedure [35,75]. The MTT assay consists of a colorimetric determination performed in microtiter plates, where absorbance measurements are obtained at the end of the assay. Its foremost assumption relies on tetrazolium as an indicator of the intracellular reducing potential, which, in turn, informs the overall cell state. Interference in formazan formation by glycolysis inhibitors has recently been shown, supporting the notion that the main source of reducing power might come from the glycolytic NAD(P) and not from mitochondria [76]. Therefore, the amount of formazan formation reflects on the intensity of aerobic glycolysis, whereas the major function of aerobic glycolysis is known to maintain high levels of glycolytic intermediates to support cellular anabolic functions [77]. Given that the amount of formazan formation depends on the number of vital cells and their metabolic activity, interpretation of results derived from MTT assays remains mainly dependent on the underlined specific experimental conditions. Therefore, we decided to use the term “cellular proliferation” when cells were stimulated with identical/fixed LPS as well as ATP concentrations, assuming that cells were identically activated and differences in the amount of formazan formation were mainly dependent on the differences in cell number, hence reflecting cell proliferation. In contrast, we used the term “cell viability” when differences in LPS and/or ATP dosages suggested that differences in the amount of formazan formation depend on the activation state and/or cell number.

Furthermore, in the case of the in vivo study, it must be mentioned that identification of the Iba1 antigen is not specific for microglia. It is rather expressed in all cells derived from a monocytic lineage, namely, resident microglia including CNS-associated macrophages (CAMs) [11] as well as migrating peripheral monocytes/macrophages [78,79]. To prove the diagnostic value of the results derived from the animal experiments presented in this study, we have to consider that the consequences of a focal stab injury in the brain involve rupture of (micro-) vessels, focal bleeding and blood–brain barrier disturbance in the surrounding brain tissue, leading to a focal neuroinflammation. Resident microglia and also resident CAMs act as the front-line cells of the immune response and—among other functions—provide for the release of chemoattractants driving the invasion of blood-borne immune-competent cells including monocytes/macrophages. The latter express Iba1, similar to the brain-resident microglia. Therefore, these cells are also Iba1-positive in the respective immunohistochemical staining.

In parallel, we noted that the expression of the Ki-67 protein is strictly associated with cell proliferation. The Ki-67 protein is present during all active phases of the cell cycle but is absent in resting cells [80,81]. In adult mice, proliferation of blood-borne monocytes/macrophages is located in the bone marrow, whereas proliferation of the resident microglia takes place within the brain tissue. Monocyte invasion into brain tissue requires disturbance of the blood–brain barrier (BBB) [82,83,84]. Recent findings revealed that ischemia-induced immune cell proliferation is strictly limited to microglia [85]. Furthermore, immediately starting immune cell expansion due to acute brain injury was mainly driven by microglial proliferation, whereas invasion of blood-borne immune cells peaked later (around five days after injury in adult mice) [15,85]. Hence, we assume that Iba1/Ki67-positive cells in brain tissue are mainly proliferating resident microglia. We determined the relation between Iba1/Ki67-positive cells and the total amount of Iba1-positive cells. Therefore, the question arose whether the results are confounded by a bias of genotype-dependent differences between the brain’s resident microglia and the blood-borne recruited monocytes/macrophages. Our data show that in wild-type mice, the proliferation rate of Iba1-positive cells in brain tissue was significantly increased compared to PI3Kγ mutant mice. There is ample evidence that leukocytes including blood-borne mono-cytes/macrophages of PI3Kγ mutant mice display a migration deficit phenotype [32,86,87,88,89]. Consequently, the number of Iba1-positive cells in brain tissue of PI3Kγ mutant mice derived from invaded blood-borne monocytes/macrophages should not be increased—more likely even reduced—compared to wild-type mice. Therefore, we assume that our finding regarding the impaired proliferation of resident microglia in mice with a loss of lipid kinase activity is reliable.

## 5. Conclusions

Collectively, our data show a key role of cellularly and mitochondrially generated ROS for adaptive responses of microglial cells to the rapid switch from a very slow propagating self-renewing phenotype to a rapidly proliferating phenotype. Furthermore, the lipid kinase activity of PI3Kγ appears to be responsible for the involvement of PI3Kγ in microglial proliferative activation and cell viability due to inflammatory stimulation.

## Figures and Tables

**Figure 1 cells-10-02534-f001:**
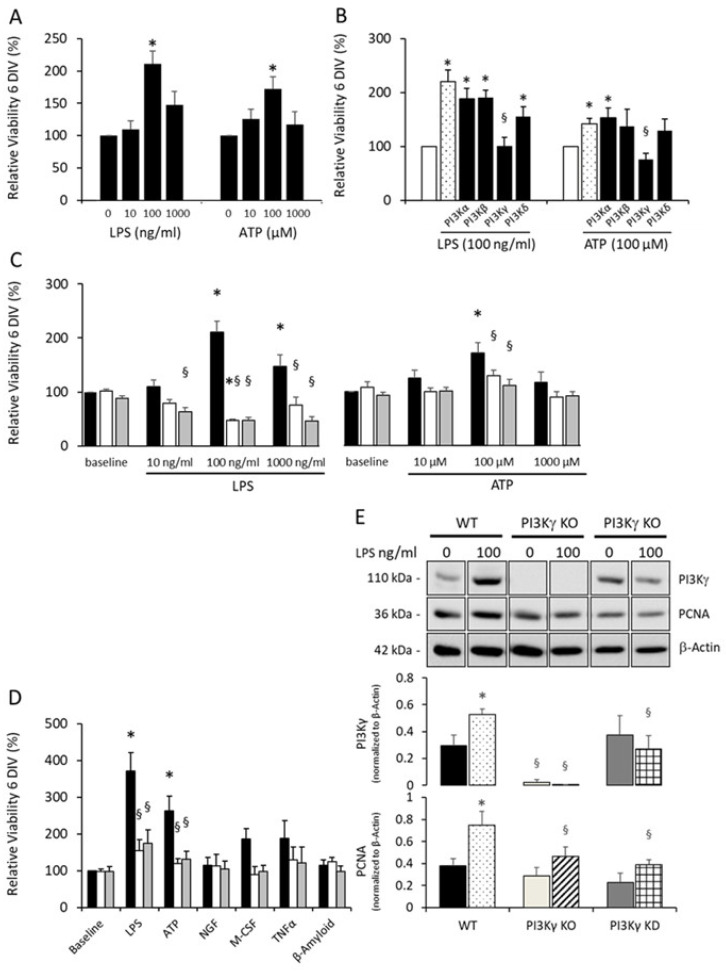
PI3Kγ mediates pattern recognition receptor-induced microglial viability and proliferation. (**A**) Microglia derived from wild-type mice (WT) were seeded (5000 cells/well) in a 96-well plate and incubated with indicated concentrations of LPS or ATP for 6 days. Proliferation was measured by using an MTT assay. *n* = 5−6, * *p* < 0.05, compared to unstimulated cells. (**B**) Primary microglia derived from wild-type mice were seeded (5000 cells/well) in a 96-well plate and pretreated for 60 min with selective inhibitors targeting lipid kinase activity of PI3Kα (A66; 300 nM), PI3Kβ (TGX221; 200 nM), PI3Kγ (AS605240; 1 µM) or PI3Kδ (IC87114; 2 µM). Subsequently, cells were incubated with LPS (100 ng/mL) or ATP (100 µM) for 6 days. Cellular viability was measured by using an MTT assay. *n* = 9, each, * *p* < 0.05, compared to untreated microglial cells (open columns); ^§^ *p* < 0.05, compared to mock 60 min-pretreated microglial cells (dotted columns). (**C**) Microglia obtained from wild-type (black columns), PI3Kγ knockout (white columns) and PI3Kγ lipid kinase-dead (gray columns) mice were seeded (5000 cells/well) in a 96-well plate and incubated with LPS or ATP at indicated concentrations for 6 days. Viability was measured by using an MTT assay. *n* = 5–6, each, * ^§^ *p* < 0.05, * significant differences vs. WT baseline, ^§^ significant differences vs. microglia derived from wild-type mice at same condition. DIV: days in vitro. (**D**) Primary microglia derived from wild type (WT, black columns), PI3Kγ knockout mice (white columns) or mice carrying a lipid kinase-dead PI3Kγ mutant (gray columns) were seeded (5000/well) in a 96-well plate and incubated with LPS (100 ng/mL), ATP (100 µM), NGF (30 ng/mL), M-CSF (30 ng/mL), TNFα (20 ng/mL) or β-amyloid (5 µM) for 6 days. Viability rate was measured by using an MTT assay. *n* = 6, each, * ^§^ *p* < 0.05; * compared to untreated wild-type microglial cells (baseline, black column), ^§^ indicates significant differences vs. WT cells with the same treatment. (**E**) Primary microglia derived from wild-type (WT), PI3Kγ knockout (KO) or PI3Kγ kinase-dead (KD) mice were seeded (100,000 cells/well) in a 6-well plate and incubated without (WT, black columns; PI3Kγ KO, white columns; PI3Kγ KD, gray columns) and with LPS (WT, dotted columns; PI3Kγ KO, hatched columns; PI3Kγ KD, grid-like columns). After 6 days, cells were harvested, lysed and separated by SDS-PAGE. Subsequent immunoblotting demonstrates cellular level of PI3Kγ, PCNA and loading control ß-actin as indicated. Upper panel, representative immunoblot. *n* = 7−11, * ^§^ *p* < 0.05, * significant differences vs. WT baseline, ^§^ significant differences vs. microglia derived from wild-type mice at same condition.

**Figure 2 cells-10-02534-f002:**
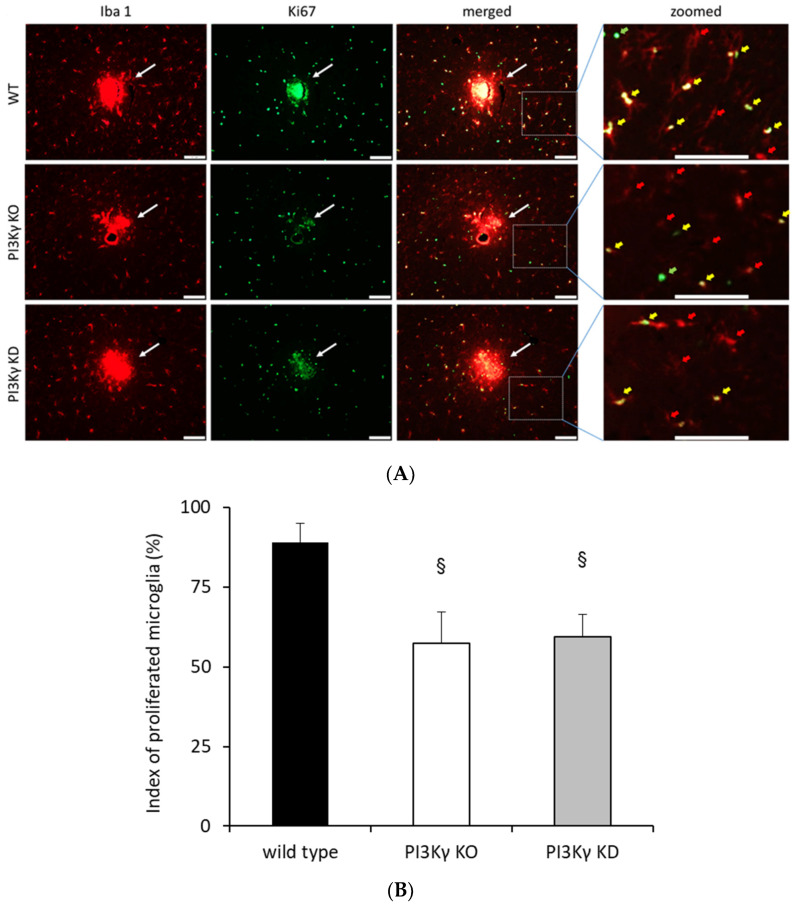
Absence of PI3Kγ lipid kinase activity reduces proliferation of microglial cells after brain injury by focal stab damage. (**A**) Representative images of proliferating Iba1-positive cells. Brain slices derived from wild-type (WT), PI3Kγ KO and PI3Kγ KD mice were co-labeled for Iba1 (cell marker for resident microglia as well as peripheral monocytes/macrophages, red, left panel) and Ki67 (proliferation marker, green, middle panel). Double-positive Iba1/Ki67 microglia show the amount of proliferated microglia (merged/zoomed pictures); yellow arrows: double-positive Iba1/Ki67 cells indicating proliferated microglia; red arrows: Iba1-positive cells indicating resident microglia as well as peripheral monocytes/macrophages; green arrows: Ki67-positive cells, indicating other proliferated cells; white arrows indicate the location of the respective focal wound injury; white asterisks indicate the center of focal stab damage. Bars indicate 100 µm. (**B**) Quantification of proliferating Iba1-positive cells. Proliferation rate of microglia in the brain of wild-type, PI3Kγ KO and PI3Kγ KD mice. Index of proliferated microglia was determined by the ratio of Iba-1/Ki67-positive cells to total number of Iba1-positive cells. *n*= 4–5 per group, ^§^ *p* < 0.05, comparison vs. brain slices derived from wild-type mice (WT).

**Figure 3 cells-10-02534-f003:**
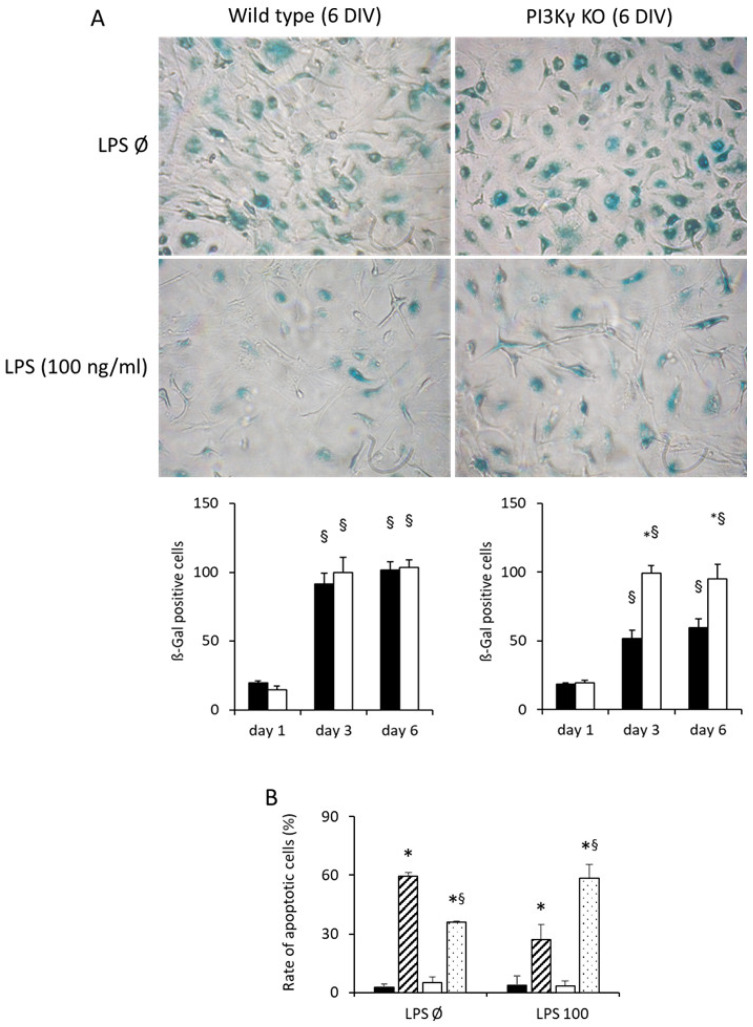
PI3Kγ suppresses induction of senescence and apoptosis of LPS-activated microglia. (**A**) Primary microglia derived from wild-type (WT, black columns) or PI3Kγ KO (white columns) mice were seeded (100,000/well) in a 12-well plate and incubated without or with LPS (100 ng/mL) for 1, 3 or 6 days, as indicated. Induction of apoptosis was monitored using senescence-associated β-galactosidase staining of cells (upper panel, representative pictures are shown for day 6). β-Gal-positive (blue) cells were counted, *n* = 3 per day and genotype, * ^§^ *p* < 0.05, * significant differences vs. WT microglial cells at the same time point, ^§^ significant differences vs. day 1 at the same genotype. (**B**) Rate of apoptotic (SubG1) cells after one or six days of in vitro incubation of microglial cells (wild-type microglia, day 1: black bars; wild-type microglia, day 6: hatched bars; KO microglia, day 1: white columns; KO microglia, day 6: dotted columns) without LPS stimulation (LPS Ø) and with LPS stimulation (LPS 100 (ng/mL)). Cells were harvested, fixed and permeabilized and subsequently analyzed for their DNA content using flow cytometry, *n*= 3 per day and genotype, * ^§^ *p* < 0.05, * significant differences between day 1 and day 6 within the same genotype, ^§^ significant differences between cells incubated without or with LPS (100 ng/mL) within the same genotype and time point. DIV: days in vitro.

**Figure 4 cells-10-02534-f004:**
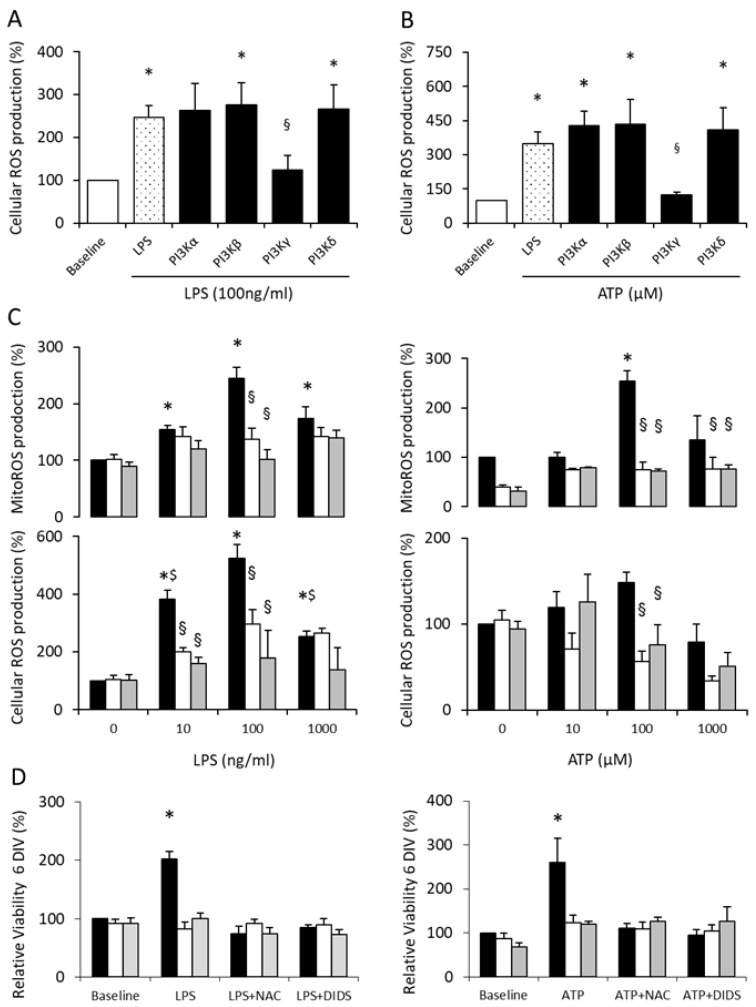
PI3Kγ controls LPS- and ATP-induced microglial ROS production. (**A**) PI3Kγ controls LPS-induced cellular ROS production in naïve microglia. Primary microglia derived from wild-type mice were seeded (30,000 cells/well) in a 96-well plate and pretreated for 60 min with selective inhibitors targeting lipid kinase activity of PI3Kα (A66; 300 nM), PI3Kβ (TGX221; 200 nM), PI3Kγ (AS605240; 1 µM) or PI3Kδ (IC87114; 2 µM). Subsequently, cells were incubated with LPS (100 ng/mL) for 6 days. Cellular ROS production was measured by DCF fluorescence. *n* = 9, each, * ^§^ *p* < 0.05, * compared to untreated microglial cells (baseline, open columns); ^§^ compared to mock pretreated microglial cells (LPS, dotted columns). (**B**) PI3Kγ controls ATP-induced cellular ROS production in naïve microglia. Primary microglia derived from wild-type mice were seeded (30,000 cells/well) in a 96-well plate and pretreated for 60 min with selective inhibitors targeting lipid kinase activity of PI3Kα (A66; 300 nM), PI3Kβ (TGX221; 200 nM), PI3Kγ (AS605240; 1 µM) or PI3Kδ (IC87114; 2 µM). Subsequently, cells were incubated with ATP (100 µM) for 6 days. Cellular ROS production was measured by DCF fluorescence. *n* = 11–12, each, * ^§^ *p* < 0.05, * compared to untreated microglial cells (baseline, open columns); ^§^ compared to mock pretreated microglial cells (LPS, dotted columns). (**C**) LPS and ATP mediate a dose-dependent stimulation of cellular (lower panel) and mitochondrial (MitoROS, upper panel) ROS production. Primary microglia derived from wild type (black bars), PI3Kγ knockout mice (white bars) or mice carrying a lipid kinase-dead PI3Kγ mutant (gray bars) were seeded (30,000 cells/well) in a 96-well plate and incubated with LPS or ATP for 6 days at the concentrations indicated. Cellular ROS production was measured by DCF fluorescence, and MitoRos production was measured by MitoSOX™ Red mitochondrial superoxide indicator. *n* = 3, each, * ^§ $^ *p* < 0.05, * significant differences from untreated WT (LPS 0, ATP 0), ^§^ significant differences from microglia derived from wild-type mice at the same condition, ^$^ significant differences from WT LPS 100 ng/mL. (**D**) PI3Kγ controls microglial proliferation via ROS. Primary microglia derived from wild type (black bars), PI3Kγ knockout mice (white bars) or mice carrying a lipid kinase-dead PI3Kγ mutant (gray bars) were seeded (5000 cells/well) in a 96-well plate and incubated with LPS (100 ng/mL) or ATP (100 µM) solely or together with NAC or DIDS for 6 days. Cellular proliferation was measured by using an MTT assay. *n* = 3, each, * *p* < 0.05, compared to untreated wild-type microglial cells (baseline, black columns). DIV: days in vitro.

## Data Availability

Data supporting the reported results can be requested from the corresponding author.

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
