# Peer review of "PI3Kγ Mediates Microglial Proliferation and Cell Viability via ROS"

_cells, 2021, doi:10.3390/cells10102534_

Round 1

Reviewer 1 Report

The manuscript by Caroline Schmidt and colleagues addresses the role of phosphoinositide 3-kinase γ  in the induction of microglia proliferation in various conditions in vitro and in vivo.

The idea that PI3K is involved in microglia proliferation and activation has been already proven by many authors however, here, the authors postulate that PI3Kγ-mediated induction of ROS production is a key factor to switch the slow proliferating resting microglia into the rapidly proliferating active cells.

This is a generally well-documented report. My main concerns refer to the manuscript organization and interpretation of some observations.

Specific comments:

1. This is very difficult to appreciate the cell morphology and antigen staining in figure 2 (which has been named with the wrong number 1 in the manuscript). The images seem not to be representative for the results they are supposed to present.

Moreover, in the case of in vivo study, it should be mentioned that identification of Iba1 antigen is not specific for microglia, it rather stains all myeloid-derived cells namely, resident microglia as well as migrating peripheral monocytes/macrophages. Particularly after injury, both populations are involved in the inflammatory response. Please refer to that.

2. The title of figure 1 said that PI3Kγ controls pattern recognition receptors-induced microglial proliferation since the figure itself shows just inhibition of cell viability and production of proliferation-associated protein in cells derived from knockout mice. It does not provide evidence that the protein of interest, in fact, controls cell proliferation and/or induction of PCNA. It seems that the title is slightly overstated in this case. The authors describe results obtained by MTT test as cell viability or cellular proliferation interchangeably. This needs to be clarified and standardize throughout the text. 3. In my opinion the manuscript could be organized in such a way that the results of PI3K effect on senescence and apoptosis are shown in the main body of the paper. Now the text describes results that are provided only in supplementary.  

4. All figures need to be properly named, marked and annotated.

Reviewer 2 Report

Schmidt and collaborators describe in the present manuscript the effect of PI3Kg mediated ROS on microglial proliferation. It is an excellent work, very clear; all the required experimentation to validate their hypothesis is presented, and the manuscript is easily followed. Just few minor comments:

- ATP dependent ROS production is clear in supp Figure 4 but not in Fig 3. Do you have any explanation?

- Starting figures with panel A will facilitate reading

- Authors claim that PCNA protein expression is increased upon prolonged incubation of WT microglia with LPS, however, no protein from untreated cells is shown in the WB. Could it be possible to have data on WT and both KO? Please specify how the immunoblot quantification is performed.

- Which is the ATP concentration used in Fig 1C and 3B? Legends seem confusing.

- Please specify what DIV stands for (Figs 1, 3, 1S, 3S)

- There are two Figure 1.

- Could it be possible to increase the image quality in figure 2?

- Please check legends of Supplemental figures, especially those parts regarding statistics and symbol description (Supp Figures 2, 3 and 4)

- Does Declaration of Helsinki apply to animal research?

Reviewer 3 Report

In this study, the authors have reported the role of PI3Kγ in microglial proliferation and apoptosis after acute inflammatory activation using LPS/ATP. The study includes important experiments to conclude the findings. It was great to see the authors used genetic manipulated / transgenic cells (KO and KD PI3Kγ cells) and PI3K pharmacological inhibitors to elucidate the role of PI3Kγ in regulating microglial proliferation and the underlying mechanism. 

However, some minor/major revisions are needed to enhance the manuscript's clarity and findings. 

1) Introduction: At several places, long sentences are there to convey the information. It is not easy to read.  E.g., Page 2 - line 45-55, line 71-75.

No need to write the line 82-83 in the introduction. It is good to end the introduction about the rationale/objective of the study.

2) Materials and Methods: 

A) Weird symbols are present. Maybe the symbols are generated automatically in the draft while editing? Line 88 (in front of actin), 103 (next to TNF), 138. 

Make sure the correct symbols are there throughout the article.

Include the catalog number of the antibodies. As we know, there are several antibodies for specific proteins are available from the same company. Hence, reporting the catalog item number always help other researchers to identify suitable antibodies to detect similar protein in their studies. 

B) It is important to be consistent with the spacing/superscript vs. subscript. E.g.,  95°C vs. 95 °C; CO2 vs. CO2

C) It is important to report fine details about the experiments. E.g., Make sure to report the density (number of cells) of microglia plated for each experiment (day 1). 

Even though the LPS and other chemical concentrations are reported in the figures and figure legend, it is important to write them in the method section. E.g., please do report the concentration of LPS or ATP in line 144. Same thing for the concentrations of the inhibitor. 

3) Result:

The authors need to reformat the figure sections. The figure order needs to be from A to C from top to bottom. Currently, it is reversed. It applies to all figures. The figure sections should be laid out from the top (starting section A) to the bottom, as you report in the description of the result section. It will be easy for readers to follow which experiment was first performed and later experiments in sequence.

Figure 1. 
The authors need to report how long (specific minutes or hours) the inhibitors were pretreated before LPS or ATP treatment. Line 235. It applies to every place where the pretreatment is mentioned.

Figure 1D - Need to have western blot lanes with WT, KO, and KD cells without and with LPS treatment. It allows understanding the normal level of PI3Kγ and PCNA expression without any stimulation and following stimulation. The PI3Kγ and PCNA band quantification data need to be reported as normalization with actin after correcting the image adding multiple lanes with and without LPS.

Line 292 - 297 can be separated section after 3.2 as it pertains to the in vivo study.

It will be worth adding the supplementary figure 1 in the actual Figure 1 on page 6 and labeling it as figure 1 section C. Doing this it will allow the readers to see in the same figure the inhibition of microglia proliferation when PI3Kγ  is blocked pharmacologically or through genetic modulation (KO/KD).

Figure2.

Figure 2 reported in line 299 is mentioned as Figure 1. 

Figure 2A - The current image only shows one microglia per image. Provide a lower magnification image with multiple microglia with the reported staining. And have the high magnification image of one microglia as an insert. It will allow the readers to see the differences between numbers of proliferating microglia in WT vs. the KO and KD brain slice culture (lower insert) with a focal wound injury. A single cell image per group is not sufficient. If possible, it will be worth having the confocal image.

Figure 3 Supplementary -  Include the supplementary figure 3 part within the main body of the result section of the manuscript. It is crucial findings, and the senescence information is stressed in the abstract and discussion.

A) In line 324, is it supposed to be Figure 3B instead of 3A?

B) Need to have a high-quality image of Figure 3 A Suppl. It will be better if the treatment information is written on the side of the image instead of on the image. 

C) Figure 3 B suppl. The graph is challenging to interpret just by looking at the image. Need to be reformatted in a way where you can label with or without LPS on the X-axis. 

Figure 3 (page 10) -  Move the current 3B section (Cellular ROS production % treatment with ATP, NGF, etc.) as supplementary data. In place of that report, the findings with ATP similar to the LPS image (section A; pharmacological inhibition of PI3K and impact on cellular ROS production % with ATP treatment).

Why the percentage LPS-mediated increase in cellular ROS production ( ~250% ; Figure 3 A section) is not similar to % increase to ~ 500% in figure 3C cellular ROS production? Why the drastic difference in cellular ROS production with LPS (100 ng/ml) treatment in different experimental setups?

4)Discussion:

Discussion is well written, connecting to the unanswered question and reported literature studies. 

It is important to discuss the findings - why at higher concentration of LPS or ATP (1000ng/ml or 1000uM respectively) - there is a decrease in viability (Figure 1A) as compared to middle dose - is it due to the toxicity aspect of LPS / ATP at a high concentration. Similarly, explanation for the decline in ROS production with high doses. Is there any link between reducing ROS production vs. decrease proliferation at higher concentrations? It can not be a toxicity-mediated reduction in cell viability as the number seems equivalent to the no treatment. 

Edit the long sentence if present. E.g., Line  428-431

Round 2

Reviewer 3 Report

Appreciate the authors for editing the draft as per recommended changes. 

However, still notice some minor corrections that need to be done.

Minor changes -

Figure 1 E - The western blot needs to be one continuous image representing one complete experiment.
Cropped blot sections are not acceptable. It gives readers an impression that the different areas of multiple blots may be cropped and put together. 
Additionally, the data need to presented as normalized to actin for the figure 1E western blot. 

Figure 2A - Don't place white asterisks mark on the center of focal stab damage. For clarity, mark the focal stab with the thin arrow. White asterisk is masking the area.

Multiple spacing issues, e.g.,  441 line for "96well". Make sure throughout the article there is no spacing issue. 

Author Response

We thank the reviewer for the positive feedback and the opportunity to improve the manuscript. We respond below point by point and made the corresponding edits in red font in the revised version of the manuscript.

Reviewer 3

Minor changes -

Figure 1 E - The western blot needs to be one continuous image representing one complete experiment.

Cropped blot sections are not acceptable. It gives readers an impression that the different areas of multiple blots may be cropped and put together.

Additionally, the data need to presented as normalized to actin for the figure 1E western blot.

Response: We added a new Figure 1 Supplement, where the western blot is shown as continuous image representing one complete experiment, from which the representative western blot image is mounted in Figure 1E.

In addition, axis label of Fig 1 E was completed as indicated.

Figure 2A - Don't place white asterisks mark on the center of focal stab damage. For clarity, mark the focal stab with the thin arrow. White asterisk is masking the area.

Response: Done.

Multiple spacing issues, e.g.,  441 line for "96well". Make sure throughout the article there is no spacing issue.

Response: Done.